# Preoperative Diagnostic Uncertainty in T2–T3 Rectal Adenomas and T1–T2 Adenocarcinomas and a Therapeutic Dilemma: Transanal Endoscopic Surgery, or Total Mesorectal Excision?

**DOI:** 10.3390/cancers13153685

**Published:** 2021-07-22

**Authors:** Xavier Serra-Aracil, Noemi Montes, Laura Mora-Lopez, Anna Serracant, Carles Pericay, Pere Rebasa, Salvador Navarro-Soto

**Affiliations:** 1Servicio de Cirugía General y del Ap. Digestivo, Departament de Cirurgia, Institut d’Investigació i Innovació Parc Tauli I3PT, Parc Tauli Hospital Universitari, Universitat Autònoma de Barcelona, 08208 Sabadell, Spain; montes.ortega@gmail.com (N.M.); mora.lopez.laura@gmail.com (L.M.-L.); annserracant@gmail.com (A.S.); pererebasa@telefonica.net (P.R.); snavarro@tauli.cat (S.N.-S.); 2Medical Oncology Department, Institut d’Investigació i Innovació Parc Tauli I3PT, Parc Tauli Hospital Universitari, Universitat Autònoma de Barcelona, 08208 Sabadell, Spain; CPericay@tauli.cat

**Keywords:** rectal adenoma, early rectal adenocarcinoma, transanal endoscopic surgery, total mesorectal excision

## Abstract

**Simple Summary:**

Endorectal ultrasound and rectal magnetic resonance are sometimes unable to differentiate between stages T2 and T3 in rectal adenomas that are possible adenocarcinomas, and between stages T1 and T2 in rectal adenocarcinomas. These cases of diagnostic uncertainty raise a therapeutic dilemma: should they be treated with transanal endoscopic surgery (TES) or total mesorectal excision (TME)? We present an observational study of a cohort of 803 patients who underwent TES between 2004 and 2021. Five hundred and twenty-nine patients operated on for adenoma (group I) and 109 for low-grade adenocarcinoma (group II) were included. Diagnosis was uncertain in 113/529 patients (21.4%) in group I, and in 8/109 (7.3%) in group II. The definitive pathology diagnosis showed 17 cases in group I (15%) to be adenocarcinomas greater than T1, and two cases in group II. On the strength of these data, in cases of diagnostic uncertainty we recommend TES as the initial indication.

**Abstract:**

Background: Endorectal ultrasound and rectal magnetic resonance are sometimes unable to differentiate between stages T2 and T3 in rectal adenomas that are possible adenocarcinomas, or between stages T1 and T2 in rectal adenocarcinomas. These cases of diagnostic uncertainty raise a therapeutic dilemma: transanal endoscopic surgery (TES) or total mesorectal excision (TME)? Methods: An observational study of a cohort of 803 patients who underwent TES from 2004 to 2021. Patients operated on for adenoma (group I) and low-grade T1 adenocarcinoma (group II) were included. The variables related to uncertain diagnosis, and to the definitive pathological diagnosis of adenocarcinoma stage higher than T1, were analyzed. Results: A total of 638 patients were included. Group I comprised 529 patients, 113 (21.4%) with uncertain diagnosis. Seventeen (15%) eventually had a pathological diagnosis of adenocarcinoma higher than T1. However, the variable diagnostic uncertainty was a risk factor for adenocarcinoma above T1 (OR 2.3, 95% CI 1.1–4.7). Group II included 109 patients, eight with uncertain diagnosis (7.3%). Two patients presented a definitive pathological diagnosis of adenocarcinoma above T1. Conclusions: On the strength of these data, we recommend TES as the initial indication in cases of diagnostic uncertainty. Multicenter studies with larger samples for both groups should now be performed to further assess this strategy of initiating treatment with TES.

## 1. Introduction

The standard treatment for rectal adenomas up to 2 cm in size is endoscopic polypectomy. Prior to the development of transanal endoscopic surgery (TES), described by Buess [1], the management of larger adenomatous lesions located in the middle and upper third of the rectum was abdominal surgery, i.e., total mesorectal excision (TME).

TME is the standard surgical treatment for rectal adenocarcinomas at stages higher than T1 [2,3]. Depending on the height of the tumor, this surgery involves a temporary or permanent ostomy. This requirement may reduce quality of life and cause genitourinary alterations [4].

Around 20% of large rectal adenomas are identified as infiltrating adenocarcinomas in the definitive pathological study of the specimen [5]. For this reason, TES with full-thickness wall excision is the treatment of choice in these tumors [6]. This technique permits excellent vision and allows complete en bloc resection with wide margins; in these conditions, the local recurrence rate is 4% [7].

In view of the high rate of adenocarcinoma in these rectal adenomas, correct staging is essential. Endorectal ultrasound (ERUS) and pelvic magnetic resonance imaging (MR) are the main diagnostic tools for establishing the in-depth diagnosis of the tumor in the wall of the rectum (T) and the possible lymph node invasion (N). ERUS has been shown to be more accurate than MR for determining the depth of the lesion, especially in the initial stage [8]. TES is the surgical treatment of choice in adenomas and in T1 adenocarcinomas that do not present the factors of poor prognosis described in the literature: namely, submucosal invasion depth sm3, poor degree of tumor differentiation, vascular invasion, lymph node invasion, perineural invasion, involvement of the resection margin (≤1 mm), lymphocyte infiltration, and tumor budding [4,9]. In “high risk” T1 and the rest of rectal adenocarcinomas, the surgery of choice is TME [3].

As neither ERUS nor MR offer the same diagnostic precision as pathological analysis [10], in some situations the diagnosis may be uncertain. This is particularly so in rectal adenomas when ERUS and/or MR give diagnoses of T2 or T3, and in rectal adenocarcinomas in which it is difficult to differentiate between T1 and T2. In these situations, which surgical approach should be preferred: TES or TME?

Deciding on the best treatment in these situations is a difficult task. There is a risk of overtreatment—that is, the performance of unnecessary major surgery, which entails greater surgical morbidity and mortality, alterations in quality of life, and higher economic costs. On the other hand, TES may prove to be insufficient for these lesions and the surgical treatment may need to be completed with TME.

Nevertheless, recently published studies in which TES has had to be followed by Completion Surgery to TME have not shown that the long-term oncological results are affected, as long as local surgery is performed in the next few weeks [5,11].

In 2004, our group established a therapeutic protocol for these lesions, based on the hypothesis that in these rectal tumors with uncertain diagnosis, the percentage requiring completion surgery to TME would not be high, and that, in any case, the final oncological results would not be worse than if TME had been performed at the outset. For this reason, in the first instance, the less-invasive surgery was proposed: i.e., TES instead of TME.

The objectives of the study, then, were to determine the frequency of uncertain diagnosis in rectal adenomas and adenocarcinomas T1 or T2; to confirm or reject our hypothesis that TES is the most appropriate treatment in these lesions of “uncertain diagnosis”, in terms of avoiding over-treatment and unnecessary TMEs; and to calculate the risk of the variable “uncertain diagnosis” as a possible predictive factor of invasive cancer higher than stage pT1.

## 2. Materials and Methods

### 2.1. Study Design

We conducted an observational cohort study of consecutive patients undergoing TES, with prospective data collection and retrospective analysis. Computerized data management was carried out using Microsoft^®^ Access 2003; data were entered in a relational database and in a protected format.

### 2.2. Patients and Setting

All patients included were operated on by surgeons from the Coloproctology Unit between June 2004 and January 2021. All rectal tumors underwent a preoperative study protocol [4]. The preoperative study was based mainly on endorectal ultrasound (ERUS) and rectal magnetic resonance imaging (MR). All patients underwent ERUS. MR was introduced in our institution for rectal tumor staging in 2007. MR for possible candidates for TES was selectively indicated in all adenocarcinomas, in cases of uncertain diagnosis, and when there was a risk of perforation into the peritoneal cavity.

Patients were classified into the following five preoperative surgical indication groups: group I with curative intent (with preoperative biopsy of adenoma), staged as (us/mr: ERUS and MR) us/mr T0–1 and us/mr N0 after ERUS and MR; group II, with curative intent (preoperative biopsy of low-grade adenocarcinoma), us/mr T0–1 and us/mr N0; group III, consensual indication (low-grade adenocarcinoma), us/mr T2 and us/mr N0, in patients who refuse radical surgery; group IV, palliative care, and group V, atypical [12].

### 2.3. Inclusion Criteria

Patients scheduled for curative TES, from groups I and II above.

### 2.4. Exclusion Criteria

Patients in groups III, IV and V above; patients who underwent direct abdominal surgery when intraoperative assessment ruled out TES.

### 2.5. Definition of Group I and II Patients with Tumors of Uncertain Diagnosis

Group I. Rectal tumors with adenoma biopsy diagnosed as T2–T3 or N1 after ERUS and/or MR. In these patients, multiple biopsies are repeated and if the pathological diagnosis continues to be adenoma, the diagnosis is established as “uncertain”.

Group II. Rectal tumors with adenocarcinoma biopsy in which ERUS suggests possible invasion beyond the submucosal layer without suspicious adenopathies (usT1–T2), N0. The diagnosis by MR is T2, N0 or lower.

### 2.6. Preoperative Preparation, Surgical Technique and Postoperative Evolution

All patients with an indication for TES underwent antegrade mechanical colon preparation together with antibiotic and thromboembolic prophylaxis according to the protocol [4]. Anesthesia was usually general, unless the anesthesiologist advised the use of spinal anesthesia. The techniques used for local excision were either TEM (Richard Wolf, Knittlingen, Germany) or transanal endoscopic operation (TEO) (Karl Storz GmbH, Tüttlingen, Germany) [13]. The tumor was removed according to the protocol, using a full-wall ultrasonic scalpel, without resection of the perirectal fat. Whenever possible, the defect was sutured without tension. The urinary catheter was removed at the end of the surgery, oral diet and ambulation were started at 6 h, and patients were discharged at 24 h except in cases of complications.

### 2.7. Main Variables

Uncertain diagnosis in group I and II patients; definitive pathology of the specimen indicating adenocarcinoma higher than T1.

### 2.8. Secondary and Other Study Variables

Epidemiological: age, sex, body mass index (BMI).

Tumor-dependent preoperative variables: ERUS, MR, distance from the proximal and distal margin of the tumor to the anal margin, tumor on the anal canal, tumor size, location by quadrant, macroscopic morphology of the tumor (flat/polypoid, sessile/ulcerated), grade of adenoma dysplasia (high or low), microscopic morphology of the biopsy (tubular, tubulo-villous, villous), TES after endoscopic polypectomy [14], pathology after TES.

The study was approved by the local Institutional Ethics Committee (CEIC: 2016-636) and complied with the criteria of the Declaration of Helsinki. STROBE guidelines for observational studies were followed.

### 2.9. Statistical Analysis

The SPSS version 26 program was used. The prospective data collection allowed analysis without the presence of missing values. In the description of the quantitative variables, the values of the mean and standard deviation are given, or the median and interquartile range (IQR) when the normality conditions were not met. The categorical variables were described in absolute numbers and percentages. The univariate statistical analysis of the quantitative variables, with independent groups, was carried out using the Student’s *t*-test, provided its conditions of application were met; otherwise the Mann–Whitney U test was applied. For categorical variables, Pearson’s X2 test or Fisher’s exact statistic was used, depending on the conditions. A value of *p* < 0.05 was considered statistically significant, with a 95% confidence interval.

Variables with statistical significance or a trend with *p* < 0.2 were entered in the multivariate analysis. Logistic regression analysis was used to obtain a predictive model able to identify the factors that predicted “adenocarcinoma higher than T1”.

## 3. Results

### 3.1. Descriptive Analysis

During the study period, 803 patients underwent TES. Distributed by preoperative surgical indication group, 529 were group I, 109 group II, 49 group III, 41 group IV, and 74 group V. Thus, 638 patients belonged to groups I and II (Figure 1). One hundred and thirteen (21.4%) patients in group I (adenoma) and eight (7.3%) in group II (pT1 adenocarcinoma) presented uncertain diagnosis. Seventeen patients (15%) in group I, and two (25%) in group II, required completion surgery for pT > 1. Two patients underwent direct abdominal surgery when intraoperative assessment ruled out TES due to the high location of the tumor.

Table 1 presents patients’ demographic characteristics and the preoperative variables of the tumors in groups I and II.

In group I, 101 (19.1%) patients had uncertain diagnosis on ERUS (usT1–2, usT > 2), and 45 (8.5%) on MR (stage higher than T2). Within this group, 17 (15%) were pT > 1.

In group II, the T1 adenocarcinoma group, eight patients (28.4%) presented uncertain diagnosis after ERUS (usT1–2). In this group, 21 (19.3%) were adenocarcinomas pT > 1.

### 3.2. ERUS and MR of the Patients with Uncertain Diagnosis in Relation to the Definitive Pathology

Table 2 describes tumors with uncertain diagnosis after ERUS and MR tests and their definitive pathological diagnosis after TES.

In group I, in the 101 patients with uncertain diagnosis using ERUS, diagnoses higher than usT1 were not always associated with the frequency of pT > 1. With diagnoses of mrT2–3 by MRI, the diagnosis was 22.2% (4 of 18 patients). In the two patients with usN1 in this adenoma group, the biopsy was repeated. In one patient, after TES, the definitive pathology was still adenoma (and was considered an inflammatory node). In the other, the definitive pathology after TES was a T1 adenocarcinoma, and so completion surgery with TME was performed.

In group II, seven of the eight patients with uncertain diagnosis who were usT1–2 on ERUS, two were pT higher than 1. Neither of the two patients with uncertain diagnosis by MR were pT > 1.

### 3.3. Univariate Analysis of Patient and Tumor Variables with Respect to the Variable “Uncertain Diagnosis”

Table 3 analyses patient and tumor variables related to the variable “uncertain diagnosis” in groups I and II.

In group I, the adenoma group, only the definitive diagnostic variable of adenocarcinoma showed statistically significant differences. In group II, because of the low number of patients with uncertain diagnosis, none of the variables presented statistically significant differences.

### 3.4. Univariate and Multivariate Analysis of Patient and Tumor Variables with Respect to the Pathological Diagnostic Variable “pT Greater Than T1”

Table 4 assesses patient and tumor variables related to the definitive pathological variable of “rectal adenocarcinoma pT stage greater than T1” in groups I and II.

In group I, the variables high-grade dysplasia, ERUS with a diagnosis higher than usT1, and uncertain diagnosis presented statistically significant associations with the risk of adenocarcinoma pT greater than T1. In group II, once again, because of the low number of patients with uncertain diagnosis, a multivariate analysis was not performed.

In the multivariate analysis in group I, all variables significant at *p* < 0.2 for the adenocarcinoma variable pT greater than T1 were included. Predictive factors were: diagnostic uncertainty (OR 2.3; 95% CI 1.1–4.7), adenoma with high-grade dysplasia (OR 2.4; 95% CI 1.1–5.2), and tumor size > 5 cm (OR 2.3; 95% CI 1.1–4.6).

## 4. Discussion

TES is a minimally invasive technique for the local excision of rectal tumors up to 20 cm from the anal margin. Its main advantages are its low morbidity and mortality and better quality of life compared to TME. In a recent study carried out by our group, the global morbidity of TES according to the Clavien–Dindo classification was 23.6% [15], with a clinically relevant morbidity rate (Clavien–Dindo > II) of 5.6%, and a mortality rate of 0.3% [16]. TES does not cause genitourinary alterations or require a temporary or permanent ostomy, and so it can be included in outpatient and same-day surgery programs [16].

Although TME is the standard treatment for rectal cancer and provides good disease control, it has several drawbacks: it may require the use of a permanent or temporary ostomy, the mortality rate ranges between 1 and 7%, and the morbidity rate is above 30%. The median hospital stay is eight days in laparoscopic surgery and nine days in open surgery [17]. More than 50% of these patients experience some form of urinary and sexual dysfunction [18,19,20]. What is more, between 30% and 55% of patients have severe symptoms of Low Anterior Resection Syndrome after completing anal preservation surgery. These symptoms may last several years and may have a major negative impact on quality of life [21,22].

Adenomatous polyps of the rectum are considered premalignant lesions with a risk of developing into adenocarcinoma. Early detection and removal are the best means to avoid the appearance of infiltrating adenocarcinoma [23]. Regarding the choice of polypectomy or TES, after endoscopic polypectomy of malignant rectal polyps with questionable margins, we propose TES with full-thickness resection; this procedure achieves disease control and obtains minimal morbidity compared with other procedures [14].

In rectal cancer, the probability of lymph node invasion in rectal tumors above T1 is high (with rates between 12 and 28% in T2, and between 36 and 79% in T3). For this reason, TES has not obtained good oncological results and the recurrence rates vary from 20% to 45% [4], and so TME is the surgery of choice.

Specific staging of rectal tumors is essential for selecting patients who may benefit from TES. Endorectal ultrasound (ERUS) and pelvic magnetic resonance imaging (MR) are the main diagnostic tools used to establish tumor depth (T) and local lymph node invasion (N). ERUS has been shown to be more accurate than MR in determining lesion depth, especially in early stage (stage I) tumors, and is considered the test of choice for selecting patients with rectal tumors for TES [8].

However, despite improvements in imaging techniques, preoperative staging does not achieve total accuracy: a variable percentage of understaged patients will require completion surgery TME [5]. The accuracy of ERUS in staging rectal cancer ranges from 63% to 96% for T and from 63% to 85% for N [24].

The tendency towards overstaging in ERUS has been attributed to the performance of preoperative biopsy and peritumoral inflammation that is difficult to distinguish from neoplastic infiltration [25,26]. This overstaging increases the risk of overtreatment in the form of unnecessary TMEs. In the literature, figures for overstaging vary widely, from 0% to 50%; a study by our group reported an overstaging rate of 10.9% [10].

For its part, understaging by ERUS has been attributed above all to stenosis of the rectum, which impedes full performance of the test. The rate of understaging ranges between 0% and 32.7%; in our study it was 8.08 [10].

MR is currently the test of choice for rectal cancer staging because it is not observer-dependent, and the images can be compared. The accuracy of T staging ranges from 65% to 86%. MR is particularly accurate for T3 and T4 tumors, with a sensitivity for predicting T3 of between 80% and 86% and a specificity between 71% and 76% [27]. However, few studies are available on MR staging of premalignant rectal tumors and initial rectal cancer due to the difficulty in distinguishing between T1 and T2 and the unclear difference between the mucosa-submucosa and muscle layers [28]. In our study protocol for rectal adeno-villous tumors, MR is not indicated in all cases; it is performed only when ERUS cannot be assessed, in the case of uncertain diagnosis, or if there is a risk of perforation of the peritoneal cavity [10,29].

This lack of complete precision with ERUS and MR creates situations of diagnostic uncertainty and presents the therapeutic dilemma of whether to perform TES or TME. We have not found a convincing answer to this question in the literature. In the present study, in group I (the adenoma group) we reported diagnostic uncertainty in 113 patients (21.4%). In 17 (15.0%) patients, the definitive pathological diagnosis was adenocarcinoma greater than T1, requiring TME completion surgery [10]. In contrast, in the certain diagnosis group, 25 patients (6%) presented adenocarcinoma greater than T1, the difference being statistically significant. No patient or tumor variables were associated with the likelihood of diagnostic uncertainty.

The logistic regression analysis in the adenoma group identified the variables uncertain diagnosis, adenoma with high-grade dysplasia, and tumor size > 5 cm as predictors of the risk of adenocarcinoma greater than T1.

With respect to the objective of the study, i.e., to establish whether these patients should undergo TES or TME, we contend that in group I patients with uncertain diagnosis, TES is the surgical technique of choice. The risk of requiring TME completion surgery is 15% (OR 2.3; 95% CI 1.1–4.7), but this means that TES is the correct approach in 85% of patients. Recent studies on TME completion surgery after TES have found that postoperative morbidity and mortality and cancer results are similar to those of patients with initial TME, provided that TME completion is performed within nine weeks of the TES [5,11].

In group II patients with uncertain diagnosis, the NCCN-2021 clinical guidelines [3] indicate TME. In the current study, the percentage of uncertain diagnosis was lower in this group than in group I, appearing in eight (7.3%) patients. Of these eight, six were pT1 and only two required TME completion surgery. Although the sample is small, it should be borne in mind that if the clinical guidelines had been applied, six of these eight patients would have undergone overtreatment in the form of an unnecessary TME.

The greatest benefits of performing TES in patients with uncertain diagnosis are the lower morbidity and mortality rates and the higher quality of life compared to TME. From the cost-benefit point of view, in an article previously published by our group [4], we calculated the cost of TES and three days of hospitalization to be €1920. Currently, the average hospital stay has been reduced to 24 h; with a hospitalization cost of €220/day, the cost of TES now would be €1480 (€1920−440). Applying this same procedure to calculate the cost of TME (surgical time + hospital stay + consumable material), with an average hospital stay of eight days and a surgical time of 220 min, we obtain an average cost of €5160: thus, the cost-benefit ratio of TES is three times lower than that of TME.

Given the relative rarity of this situation, multicenter studies with larger samples for both groups should be performed to further assess this therapeutic strategy of initiating treatment with TES, in the knowledge that TME completion surgery may eventually be required [5,11].

The main limitation of the study is its retrospective design, although the prospective introduction of the data ensured that no cases were lost. The sample in the present study is insufficient to be able to establish a definitive conclusion. However, the trend recorded argues in favor of the indication of TES as first choice, given that this strategy avoids unnecessary overtreatment.

Its main strength is that it is a single-center study applying a homogeneous approach in a large sample. The implementation of colorectal cancer screening programs means that situations of this kind are occurring more frequently and require an effective response [7].

## 5. Conclusions

In the case of a rectal adenovillous tumor with uncertain diagnosis, despite the increased risk of a definitive pathological result of an adenocarcinoma greater than pT1, we suggest TES as the initial indication. In the case of uncertain diagnosis in a rectal adenocarcinoma between T1–2, N0, the sample in the present study is insufficient to be able to establish a definitive conclusion. Multicenter studies with larger samples of T2–T3 rectal adenomas and T1–T2 adenocarcinomas with uncertain diagnosis should now be performed to further assess this therapeutic strategy.

## Figures and Tables

**Figure 1 cancers-13-03685-f001:**
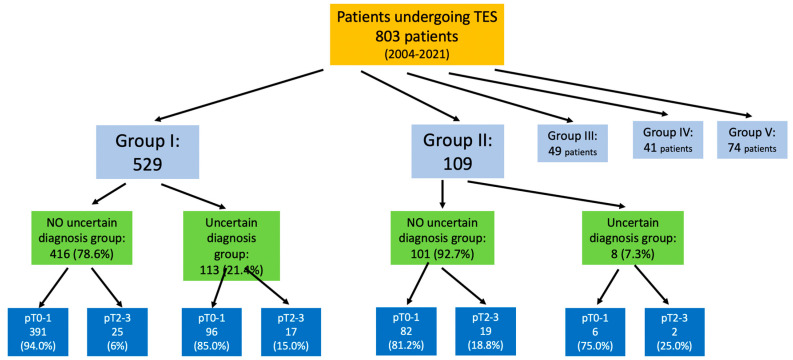
Flowchart of the study patients and those with uncertain diagnosis in groups I and II.

**Table 1 cancers-13-03685-t001:** Patients’ demographic characteristics and preoperative data of tumors in groups I and II.

Variable		Group I (n = 529)	Group II (n = 109)
Sex (%)	Female	210 (39.7%)	53 (48.6%)
	Male	319 (60.3%)	56 (51.4%)
Age (years). Median (IQR ^; range)		70 (17; 31–91)	69 (15; 41–91)
BMI (Kg/m^2^) *	<30	411 (77.7%)	69 (63.3%)
	≥30	96 (18.1%)	36 (33.0%)
Distance from lower margin to anal verge (cm). Median (IQR ^; range)		7 (5; 0–22)	8 (5; 2–22)
Tumor above anal canal	Tumors above anal canal (>4 cm)	376 (71.1%)	89 (81.7%)
	Tumors above anal canal (≤4 cm)	153 (28.9%)	20 (18.3%)
Distance to anal verge	Distance to anal verge < 6 cm	245 (46.3%)	35 (32.1%)
	Distance to anal verge between 6–12 cm	252 (47.6%)	60 (55.0%)
	Distance to anal verge > 12 cm	32 (6%)	14 (12.8%)
Size (cm). Median (IQR ^; range)		4 (2; 1–12)	3 (2; 1–7)
Tumor size	Small (≤5 cm)	321 (60.7%)	75 (68.8%)
	Large (>5 cm)	208 (39.3%)	2 (1.8%)
Distance upper margin to anal verge (cm). Median (IQR ^; range)		11 (5; 3–25,5)	11 (4; 5–26)
Tumors with upper margin 15 cm from anal verge	<15 cm	424 (80.2%)	94 (86.2%)
	≥15 cm	105 (19.8%)	15 (13.8%)
Tumor location by quadrant	Anterior	144 (27.2%)	29 (26.6%)
	Lateral	238 (45.0%)	47 (43.1%)
	Posterior	147 (27.8%)	33 (30.3%)
Lesion morphology	Flat/polypoid	117 (22.1%)	44 (41.1%)
	Sessile/ulcerated	272 (51.4%)	34 (31.8%)
Adenoma dysplasia (n = 527) *	Low grade	233 (44.2%)	
	High grade	294 (55.8%)	
Adenoma morphology (n = 523) *	Tubular	38 (7.3%)	
	Tubular-villous	231 (44.2%)	
	Villous	254 (48.6%)	
Endorectal ultrasound (us)	usT0, usT1	400 (75.6%)	84 (77.1%)
	usT1–2,2,3.	101 (19.1%)	7 (6.4%)
	Non-evaluable	28 (5.3%)	18 (16.5%)
Rectal magnetic resonance imaging (mr)	T0,1,2	232 (43.9%)	42 (95.5%)
	T > 2	45 (8.5%)	2 (4.5%)
Uncertain diagnosis (us/mr)	No	416 (78.6%)	101 (92.7%)
	Uncertain diagnosis (us/mr)	113 (21.4%)	8 (7.3%)
TES after polypectomy	No	518 (97.9%)	68 (64.8%)
	Yes	11 (2.1%)	37 (35.2%)
Pathology after TES	Adenoma	420 (79.4%)	2 (1.9%)
	Adenocarcinoma	104 (19.7%)	66 (61.1%)
	^$^ No pathology	5 (0.9%)	40 (37.0%)
pT > 1	pT0–1	487 (92.1%)	88 (80.7%)
	pT > 1	42 (7.9%)	21 (19.3%)

BMI: Body mass index. ^ IQR: Interquartile range. ^$^ No pathology after post-polypectomy TES. * The missing data correspond to patients referred to our hospital without this information.

**Table 2 cancers-13-03685-t002:** Uncertain diagnosis of groups I and II based on ERUS or MR.

Variable		Group I (n = 113)				Group II (n = 8)		
		Total	Adenomas	pT1	pT > 1	Total	No Pathology/pT1	pT > 1
Rectal ultrasound (us)	usT1–2	24	17 (70.8%)	2 (8.3%)	5 (20.8%)	7	5 (71.4%)	2 (28.6%)
	usT2	61	39 (63.9%)	16 (26.2%)	6 (9.8%)			
	usT3	16	8 (50%)	4 (25%)	4 (25%)			
	usN1	2	1 (50%)	1 (50%)	0 (0%)			
Rectal magnetic resonance imaging (mr)	mrT2–3	18	10 (55.6%)	4 (22.2%)	4 (22.2%)	2	2	0

**Table 3 cancers-13-03685-t003:** Analysis of group I and II patients regarding diagnostic uncertainty.

Variable		Group I (n = 529)			Group II (n = 109)		
		No Uncertain Diagnosis (n:416)	Uncertain Diagnosis (n:113)	*p*	No Uncertain Diagnosis (n:101)	Uncertain Diagnosis (n:8)	*p*
Sex (%)	Female	173 (41.6%)	37 (32.7%)	0.104	50 (49.5%)	3 (37.5%)	0.717
	Male	243 (58.4%)	76 (67.3%)		51 (50.5%)	5 (62.5%)	
Age (years). Median (IQR; range)		70 (17; 31–91)	71 (17; 40–89)	0.184	68 (15; 41–91)	74.5 (24; 45–82)	0.569
BMI (Kg/m^2^)	<30	324 (81.2%)	87 (80.6%	0.890	62 (63.9%)	7 (87.5%)	0.259
	≥30	75 (18.8%)	21 (19.4%)		35 (36.1%)	1 (12.5%)	
Distance from lower margin to anal verge (cm). Median (IQR; range)		7 (6; 1–22)	7 (5; 0–18)	0.977	8 (5; 2–22)	7.5 (5; 4–15)	0.704
Tumor above anal canal	Tumors above anal canal (>4 cm)	292 (70.2%)	84 (74.3%)	0.415	83 (82.2%)	6 (75.0%)	0.637
	Tumors above anal canal (≤4 cm)	124 (29.8%)	29 (25.7%)		18 (17.8%)	2 (25.0%)	
Distance to anal verge	Distance to anal verge < 6 cm	192 (46.2%)	53 (46.9%)	0.715	32 (31.7%)	3 (37.5%)	0.942
	Distance to anal verge entre 6–12 cm	197 (47.4%)	55 (48.7%)		56 (55.4%)	4 (50.0%)	
	Distance to anal verge > 12 cm	27 (6.5%)	5 (4.4%)		13 (12.9%)	1 (12.5%)	
Size (cm). Median (IQR; range)		4 (2; 1–12)	4 (2; 1–12)	0.096	2.5 (2; 1–7)	3.9 (2; 1–6)	0.018
Tumor size > 5 cm	Small (≤5 cm)	260 (62.5%)	61 (54.0%)	0.105	95 (94.1%)	5 (62.5%)	0.018
	Large (>5 cm)	156 (37.5%)	52 (46.0%)		6 (5.9%)	3 (37.5%)	
Distance from upper margin to anal verge (cm). Median (IQR; range)		11 (5; 3–25)	11 (4.5; 4–21)	0.654	10.5 (4.5; 5–26)	10.5 (3.7; 7.5–18)	0.503
Tumors with upper margin 15 cm from anal verge	<15 cm	335 (80.5%%)	89 (78.8%)	0.691	87 (86.1%)	7 (87.5%)	1
	≥15 cm	81 (19.5%)	24 (21.2%)		14 (13.9%)	1 (12.5%)	
Tumor location by quadrant	Anterior	110 (26.4%)	34 (30.1%)	0.637	27 (26.7%)	2 (25.0%)	0.913
	Lateral	187 (45.0%)	51 (45.1%)		43 (42.6%)	4 (50.0%)	
	Posterior	119 (28.6%%)	28 (24.8%)		31 (30.7%)	2 (25.0%)	
Lesion morphology	Flat/polypoid	198 (47.9%)	48 (42.5%)	0.339	57 (57.6%)	3 (37.5%)	0.295
	Sessile/ulcerated	215 (52.1%)	65 (57.5%)		42 (42.4%)	5 (62.5%)	
Adenoma dysplasia (n = 527)	Low grade	194 (46.9%)	39 (34.5%)	0.339			
	High grade	220 (53.1%)	74 (65.5%)				
Adenoma morphology (n = 523)	Tubular/Tubular-villous	217 (52.8%)	52 (46.4%)	0.242			
	Villous	194 (47.2%)	60 (53.6%)				
TES after endoscopic polypectomy	No	405 (97.4%)	113 (100%)	0.131	61 (62.9%)	7 (87.5%)	0.255
	Yes	11 (2.7%)	0 (0%)		36 (37.1%)	1 (12.5%)	
Pathology after TES	Adenoma	349 (83.9%)	70 (61.9%)	<0.001	2 (2%)	0 (0%)	0.288
	Adenocarcinoma	63 (15.1%)	42 (37.2%)		60 (59.4%)	7 (87.5%)	
	^$^ No pathology	4 (1%)	1 (0.9%)		39 (38.6%)	1 (12.5%)	
pT	pT0–1	391 (94.0%)	96 (85.0%)	0.003	82 (81.2%)	6 (75.0%	0.649
	pT > 1	25 (6%)	17 (15.0%)		19 (18.8%)	2 (25.0%	

BMI: Body mass index. IQR: Interquartile range. ^$^ No pathology after post-polypectomy TES. pT: Tumor stage (T) after TES.

**Table 4 cancers-13-03685-t004:** Univariate analysis of patients in groups I and II regarding the pathological variable of rectal cancer T2–3.

Variable		Group I (n = 529)			Group II (n = 109)		
		pT0–1 (n:487)	pT > 1 (n:42)	*p*	pT0–1 (n:88)	pT > 1 (n:21)	*p*
Sex (%)	Female	195 (40%)	15 (35.7%)	0.625	45 (51.1%)	8 (38.1%)	0.336
	Male	292 (60%)	27 (64.3%)		43 (48.9%)	13 (61.9%)	
Age (years). Median (IQR; range)		70 (16; 31–91)	67 (23; 33–87)	0.827	68.5 (17; 45–91)	70 (14; 41–84)	0.564
BMI (Kg/m^2^)	<30	382 (82.2%)	29 (69%)	0.061	52 (61.9%)	17 (81%)	0.126
	≥30	83 (17.8%)	13 (31%)		32 (38.1%)	4 (19%)	
Distance from lower margin to anal verge (cm). Median (IQR; range)		7 (5; 0–22)	7.5 (6; 2–12)	0.603	8 (4; 3–19)	6 (4; 2–22)	0.011
Tumor above anal canal	Tumors above anal canal (>4 cm)	346 (71%)	30 (71.4%)	1	75 (85.2%)	14 (66.7%)	0.062
	Tumors above anal canal (<4 cm)	141 (29%)	12 (28.6%)		13 (14.8%)	7 (33.3%)	
Distance to anal verge	Distance to anal verge < 6 cm	230 (47.2%)	16 (38.1%)	0.026 ^Φ^	25 (28.4%)	10 (47.6%)	0.171
	Distance to anal verge > 6 cm	257 (52.8%)	26 (61.9%)		63 (51.6%)	11 (52.4%)	
Size (cm). Median (IQR ^; range)		4 (2; 1–12)	5 (3; 1–11)	0.168	2 (3; 1–6)	4 (3; 2–7)	<0.001
Tumor size	Small (≤5 cm)	301 (61.8%)	20 (47.6%)	0.099	85 (96.6%)	15 (71.4%)	0.002
	Large (>5 cm)	186 (38.2%)	22 (52.4%)		3 (3.4%)	6 (28.6%)	
Distance from upper margin to anal verge (cm). Median (IQR; range)		11 (5,1; 3–25)	12 (4.1; 6–17)	0.264	10.75 (4; 5–21)	11 (3.9; 5–26)	0.893
Tumors with upper margin 15 cm from anal verge	<15 cm	390 (80.1%)	34 (81%)	1	75 (85.2%)	19 (90.5%)	0.731
	>15 cm	97(19.9%%)	8 (19%)		13 (14.8%)	2 (9.5%)	
Tumor location by quadrant	Anterior	132 (27.1%)	12 (28.6%)	0.399	20 (22.7%)	9 (42.9%)	0.143
	Lateral	216 (44.4%)	22 (52.4%)		41 (46.6%)	6 (28.6%)	
	Posterior	139 (28.5%)	8 (19%)		27 (30.7%)	6 (28.6%)	
Lesion morphology	Flat/polypoid	231 (47.7%)	15 (35.7%)	0.149	57 (66.3%)	3 (14.3%)	<0.001
	Sessil/ulcerated	253 (53.3%)	27 (64.3%)		29 (33.7%)	18 (85.7%)	
Adenoma dysplasia (n = 527)	Low grade	223 (46%)	10 (23.8%)	0.006			
	High grade	262 (54%)	32 (76.2%)				
Adenoma morphology (n = 523)	Tubular/Tubular-villous	251 (52.1%)	18 (43.9%)	0.333			
	Villous	231 (47.9%)	23 (56.1%)				
Rectal ultrasound (us)	usT0, usT1	375 (77%)	25 (59.5%)	0.015			
	usT1–2,2,3. Not evaluable	112 (23%)	17 (40.5%)				
Rectal magnetic resonance imaging (mr)	T0,1,2	214 (84.6%)	18 (75%)	0.245			
	T > 2	39 (15.4%)	6 (25%)				
Uncertain diagnosis (us/mr)	No	391 (80.3%)	25 (59.5%)	0.003	82 (93.2%)	19 (90.5%)	0.649
	Uncertain diagnosis (us/mr)	96 (19.7%)	17 (40.5%)		6 (6.8%)	2 (9.5%)	
TES after endoscopic polypectomy	NO	470 (98.1%)	35 (94.6%)	0.183	48 (56.5%)	20 (100%)	<0.001
	SI	9 (1.9%)	2 (5.4%)		37 (43.5%)	0 (0%)	

BMI: Body mass index. ^ IQR: Interquartile range. pT: Tumor stage (T) after TES. pT > 1: Group of patients with pathological/pathology study diagnosis of adenocarcinoma higher than T1 (pT > 1). ^Φ^ Likelihood Ratio.

## Data Availability

The computerized data management was carried out using Microsoft^®^ Access 2003. The data were entered in a relational database and in a protected format.

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
