# Peer review of "Preoperative Diagnostic Uncertainty in T2–T3 Rectal Adenomas and T1–T2 Adenocarcinomas and a Therapeutic Dilemma: Transanal Endoscopic Surgery, or Total Mesorectal Excision?"

_cancers, 2021, doi:10.3390/cancers13153685_

Round 1
Reviewer 1 Report
I greatly appreciate the opportunity to review the manuscript titled “Preoperative diagnostic uncertainty in T2-T3 rectal adneomas and T1-T2 adenocarcinomas and a therapeutic dilemma: Transanal Endoscopic Surgery, or Total Mesorectal Excision”. They analyzed a prospective database based on transanal endoscpic surgery for rectal adenomas/early adenocarcinomas at a single institution. They found that the risk of understaging is limited and that TES should be offered to early stage adenocarcinomas and adenomas given that understaging was limited. The authors hasve to be congratulated on this rigorous data collection. Finding new strategies to limit morbidity and mortality in patients with rectal disease is important, TES is definitely one of such strategies. I though have a couple of comments:
- Simple Summary: Please rewrite so that the conclusion is the result of the text above.
- Abstract: Please rewrite the conclusion – based on observational data, such a strict conclusion cannot be drawn. This would have to be confirmed by other studies
- Introduction: Please specify for which patients TES is an option in T1 adenocarcinomas – there are further characteristics that should be checked prior to TES (e.g. nodal status,…)
- Methods: please say what usmr means as abbreviation
- Methods: how many patients underwent direct abdominal surgery when intraop assessment ruled out TES? What were the reasons for such a ruling out?
- I think it needs further clarification of when a tumor was assessed to be uncertain by diagnostic tools or not. When was the decision made to perform EUS and MRI, when just one of them?
- How were patients treated with TES but suspicious adenopathies? Nodes were not resected with TES, correct?
- Table 1: please provide the number of eligible patients (e.g. MRI) or indicate how many did not have a certain diagnostic
- How comes that 40 patients of group 2 did not have a pathology report after TES? Was the specimen not sent to pathology or was no tumor left in the specimen?
- I don't think it is a fair point mentioning that just a small number of patients is clinically not relevant (e.g. group II), this should be corrected
- If patients were treated with TES and showed pT1 cancer – was follow-up different given that there is a slightly higher risk of lymph node metastasis? How did those patients undergo f/u – was a local recurrence found at one point?
- Please weaken the conclusion also at the end of the manuscript. 1. Based on the results of this study, one cannot say that the initial indication IS TES, I think this should be an option but not yet Evidence Level 1 recommendation. 2. The data do not allow to draw the conclusion that TME has same outcomes if performed a priori or after TES.
- Last paragraph: this is a limitation – not a conclusion
Author Response
Cancers Journal
Reviewer 1
Comments from authors:
Thank you for your very thorough comments, in response to which we have carried out a revision of our paper
Comments and Suggestions for Authors
I greatly appreciate the opportunity to review the manuscript titled “Preoperative diagnostic uncertainty in T2-T3 rectal adenomas and T1-T2 adenocarcinomas and a therapeutic dilemma: Transanal Endoscopic Surgery, or Total Mesorectal Excision”. They analyzed a prospective database based on transanal endoscopic surgery for rectal adenomas/early adenocarcinomas at a single institution. They found that the risk of understaging is limited and that TES should be offered to early stage adenocarcinomas and adenomas given that understaging was limited.
The authors have to be congratulated on this rigorous data collection. Finding new strategies to limit morbidity and mortality in patients with rectal disease is important, TES is definitely one of such strategies. I though have a couple of comments:
- Simple Summary: Please rewrite so that the conclusion is the result of the text above.
Thank you for this comment, we have removed the last sentence and completed the previous paragraph as follows:
“Observational study of a cohort of 803 patients who underwent TES between 2004 and 2021. Five hundred and twenty-nine patients operated on for adenoma (group I) and 109 for low-grade adenocarcinoma (group II) were included. Diagnosis was uncertain in 113/529 patients (21.4%) in group I, and in 8/109 (7.3%) in group II. The definitive pathology diagnosis showed 17 cases in group I (15%) to be adenocarcinomas greater than T1, and two cases in group II. On the strength of these data, in cases of diagnostic uncertainty we recommend TES as the initial indication.”
- Abstract: Please rewrite the conclusion – based on observational data, such a strict conclusion cannot be drawn. This would have to be confirmed by other studies
We have rewritten the conclusion, based on your suggestion:
Conclusions:
“On the strength of these data, in cases of diagnostic uncertainty we recommend TES as the initial indication. However, multicenter studies with larger samples for both groups should now be performed to further assess this therapeutic strategy of initiating treatment with TES”
- Introduction: Please specify for which patients TES is an option in T1 adenocarcinomas – there are further characteristics that should be checked prior to TES (e.g. nodal status,…)
We have rewritten the paragraph adding the indications for TES in T1 rectal adenocarcinoma without poor prognosis factors and in T1 rectal adenocarcinomas with indication of TME when any of the poor prognosis factors are observed (with a new reference):
TES is the surgical treatment of choice for adenomas and in T1 adenocarcinomas that do not present the factors of poor prognosis described in the literature: namely, submucosal invasion depth sm3, poor degree of tumor differentiation, vascular invasion, lymph node invasion, perineural invasion, involvement of the resection margin (≤ 1 mm), lymphocyte infiltration, and tumor budding [4,9]. In the latest “high risk” T1 and the rest of rectal adenocarcinomas the surgery of choice is TME [3].
- Methods: please say what usmr means as abbreviation
On page 6, line 130 the abbreviation is explained: (usmr : ERUS and MRI). I have corrected MRI to MR, because magnetic resonance imaging appears abbreviated as MR throughout the text.
- Methods: how many patients underwent direct abdominal surgery when intraop assessment ruled out TES? What were the reasons for such a ruling out?
On page 8, in the results section, we have added the following sentence:
“Two patients underwent direct abdominal surgery when intraoperative assessment ruled out TES due to the high location of the tumor.”
- I think it needs further clarification of when a tumor was assessed to be uncertain by diagnostic tools or not. When was the decision made to perform EUS and MRI, when just one of them?
We have added the following sentences in the manuscript:
The preoperative study was based mainly on endorectal ultrasound (ERUS) and rectal magnetic resonance imaging (MR). All patients underwent ERUS. MR was introduced in our institution for rectal tumor staging in 2007. MR for possible candidates for TES was selectively indicated in all adenocarcinomas tumors, in cases of uncertain diagnosis, and when there was a risk of perforation into peritoneal cavity..
- How were patients treated with TES but suspicious adenopathies? Nodes were not resected with TES, correct?
Those two patients had adenoma rectal tumors (table2). The biopsy was repeated. In one patient after TES the definitive pathology was still adenoma (and was considered an inflammatory node). In the other, the definitive pathology after TES was a T1 adenocarcinoma, and so completion surgery with TME was performed.
- Table 1: please provide the number of eligible patients (e.g. MRI) or indicate how many did not have a certain diagnostic
Tables 1 and 3 are complementary tables with information on patients’ demographic characteristics and preoperative data of tumors in groups I and II, and the analysis of group I and II patients regarding diagnostic uncertainty
- How comes that 40 patients of group 2 did not have a pathology report after TES? Was the specimen not sent to pathology or was no tumor left in the specimen?
You are right: as explained in table 3, these patients did not present pathology after post-polypectomy TES. We discussed these cases in a previous study (reference 14):
- I don't think it is a fair point mentioning that just a small number of patients is clinically not relevant (e.g. group II), this should be corrected
Thank you for this observation. We have changed the sentence starting on line 230 to:” In the T1 adenocarcinoma group, despite the low number of patients with uncertain diagnosis, none of the variables presented statistically significant differences”. On line 245 we have also changed the sentence: “In group II, again, because of low number of patients with uncertain diagnosis, a multivariate analysis was not performed.”
- If patients were treated with TES and showed pT1 cancer – was follow-up different given that there is a slightly higher risk of lymph node metastasis? How did those patients undergo f/u – was a local recurrence found at one point?
The definitive pathology marks the prognosis of all patients. This same criterion was applied to patients with uncertain criteria. All those with poor prognostic factors underwent MTE surgery. Patients with pathologies with a good prognosis underwent the follow-up protocol, during which no differences were observed.
- Please weaken the conclusion also at the end of the manuscript. 1. Based on the results of this study, one cannot say that the initial indication IS TES, I think this should be an option but not yet Evidence Level 1 recommendation. 2. The data do not allow to draw the conclusion that TME has same outcomes if performed a priori or after TES.
We have qualified the conclusion at the end of the manuscript:
Diagnosis was uncertain in 113 patients (21.4%) in the rectal adenoma group (group I), and in eight patients (7.3%) in the pT1 adenocarcinomas group (group II). TME completion surgery was required due to pT> 1 in 17 patients (15%) in group I and in two patients (25%) in group II.
In the case of a rectal adenovillous tumor with uncertain diagnosis, in spite of the increased risk of a definitive pathological result of an adenocarcinoma greater than pT1, we suggest TES as the initial indication. In the case of uncertain diagnosis in a rectal adenocarcinoma between T1-2, N0, the sample in the present study is insufficient to be able to establish a definitive conclusion. Multicenter studies with larger samples for both groups should now be performed to further assess this therapeutic strategy
- Last paragraph: this is a limitation – not a conclusion
We have includes this last paragraph in the limitation section:
The main limitation of the study is its retrospective design, although the prospective introduction of the data ensured that no cases were lost. The sample in the present study is insufficient to be able to establish a definitive conclusion. However, the trend recorded argues in favor of the indication of TES as first choice, given that this strategy avoids unnecessary overtreatment.

Reviewer 2 Report
The aim of this article is interesting, involving if rectal ademoma/adenocarcinoma with uncertain diagnosis/staging should be treated by transanal endoscopic surgery (TES) or by major surgery (TME). The study is rich of data and the conclusion appropriate. However, the postulate of this study is that TME should be reserved forT2 or more and that TES could be applied for T1. A more accurate pathology study could be identificate in the group of T1, advanced subgroups -sm2-3 or a submucosa involvement>1mm- in which the potential node involvement is similar to T2 (10-20%). These "high risk" T1 can safely be treated by TES or major surgery is required?
Author Response

(The authors gave the same response as above.)

Reviewer 3 Report
The abstract and simple summary however do not actually state the outcomes of the study. Some summary of the group 1 results can be made here.
In table 4, the percentages should be within the groups, not across the groups as these numbers make no sense. For example, for group 1 (PT0-1), 40% of the patients are female and 60% are male, where for group 1 (P>1), 36% female, 64% male.
Author Response
Cancers Journal
Reviewer 3
Comments from authors:
Thank you for your very thorough comments, in response to which we have carried out a revision of our paper
Comments and Suggestions for Authors
The abstract and simple summary however do not actually state the outcomes of the study. Some summary of the group 1 results can be made here.
With limits of 200 words for the abstract and 150 words for the simple summary, it is difficult to describe all the outcomes of interest.
In the simple summary all the outcomes in chart study (figure 1) are included. There are currently 145 words.
The same is true of the abstract, in which we describe the main results with the multivariate analysis of group I.
If we included more results here it would be difficult explain the aim of the study, the methodology, and the conclusions.
In table 4, the percentages should be within the groups, not across the groups as these numbers make no sense. For example, for group 1 (PT0-1), 40% of the patients are female and 60% are male, where for group 1 (P>1), 36% female, 64% male.
We have made the changes within the groups as you suggested

Reviewer 4 Report
This paper examines the possible use of less aggressive TES for treatment of uncertain diagnosis cases in rectal adenoma/adenocarcinoma. I find the paper basically fine.
Is there some way to mention the use of category of uncertain diagnosis in other institutions. For example, indeterminate colitis is not tolerated as a diagnosis in some places and provisionally accepted in some places.
In terms of presentation, in Tables 3 and 4, I suggest putting the key data (pT in Table 3; bottom box)(uncertain diagnosis in Table 4; second from bottom) in bold.
In Table 3, line on Tumor Size has the lesser than or equal sign reversed in the small category.
In the discussion, I would like to see a comment on cost to patients for TME following TES and cost benefits for TES vs. TME in this case.
Finally, please add a line to explain, polypectomy vs. TES.
Author Response
Cancers Journal
Reviewer 4
Comments from authors:
Thank you for your very thorough comments, in response to which we have carried out a revision of our paper
Comments and Suggestions for Authors
This paper examines the possible use of less aggressive TES for treatment of uncertain diagnosis cases in rectal adenoma/adenocarcinoma. I find the paper basically fine.
- Is there some way to mention the use of category of uncertain diagnosis in other institutions. For example, indeterminate colitis is not tolerated as a diagnosis in some places and provisionally accepted in some places.
We have not found this definition or topic in the literature. We chose this term because we consider it to be the most appropriate.
- In terms of presentation, in Tables 3 and 4, I suggest putting the key data (pT in Table 3; bottom box)(uncertain diagnosis in Table 4; second from bottom) in bold.
These key data in tables 3 and 4 are in bold, as you suggested.
- In Table 3, line on Tumor Size has the lesser than or equal sign reversed in the small category.
Thank you for pointing out this error. We have changed the sign to “≤”.
- In the discussion, I would like to see a comment on cost to patients for TME following TES and cost benefits for TES vs. TME in this case.
We have added the sentences below at the end of the discussion:
The greatest benefits of performing TES in patients with uncertain diagnosis are the lower morbidity and mortality rates and the higher quality of life compared to TME. From the cost-benefit point of view, in an article previously published by our group [4], we calculated the cost of TES and three days of hospitalization to be € 1920. Currently, the average hospital stay has been reduced to 24 hours; with a hospitalization cost of € 220 / day, the cost of TES now would be € 1,480 (€ 1,920 -440). Applying this same procedure to calculate the cost of TME (surgical time + hospital stay + consumable material), with an average hospital stay of eight days and a surgical time of 220 minutes, we obtain an average cost of € 5,160: thus, the cost-benefit ratio of TES is three times lower than that of TME.
- Finally, please add a line to explain, polypectomy vs. TES.
We have included the following sentence in the discussion:
Regarding the choice of polypectomy or TES, after endoscopic polypectomy of malignant rectal polyps with questionable margins, we propose TES with full-thickness resection; this procedure achieves disease control and obtains minimal morbidity compared with other procedures [14].

Round 2
Reviewer 1 Report
Thanks for revising the manuscript. It’s scientific impact improved. I still have a couple of issues:
- 5. Conclusions: please delete the first part, this is results, not conclusion
- 5. Conclusions: In the case of… please indicate at the end of the sentence WHY you think this should be the right way to do. Should it be despite instead of in spite?
- 5. Conclusions: Last two sentences. I don’t understand what the authors want to say. What should a reader do in those situations?
- Abbreviation usmr: sorry, this abbreviation is not very intuitive… would consider to change – in the table it is us/mr
-How were patients treated with TES but suspicious adenopathies? Nodes were not resected with TES, correct?
Those two patients had adenoma rectal tumors (table2). The biopsy was repeated. In one patient after TES the definitive pathology was still adenoma (and was considered an inflammatory node). In the other, the definitive pathology after TES was a T1 adenocarcinoma, and so completion surgery with TME was performed.
- I think that this has to be mentioned in the text!
- Table 1: please provide the number of eligible patients (e.g. MRI) or indicate how many did not have a certain diagnostic
Tables 1 and 3 are complementary tables with information on patients’ demographic characteristics and preoperative data of tumors in groups I and II, and the analysis of group I and II patients regarding diagnostic uncertainty
- if not all patients have had a MR exam, this has to be mentioned. Same e.g. for TES after polypectomy, this does not sum up to 529. What happened to the remaining 13 patients? Same for lesion morphology… not summing up numbers is very confusing
- How comes that 40 patients of group 2 did not have a pathology report after TES? Was the specimen not sent to pathology or was no tumor left in the specimen?
You are right: as explained in table 3, these patients did not present pathology after post-polypectomy TES. We discussed these cases in a previous study (reference 14):
- This manuscript has the focus final pathology results. How can you keep patients in the analysis where you don’t have a final pathology report? Please comment.
- I don't think it is a fair point mentioning that just a small number of patients is clinically not relevant (e.g. group II), this should be corrected
Thank you for this observation. We have changed the sentence starting on line 230 to:” In the T1 adenocarcinoma group, despite the low number of patients with uncertain diagnosis, none of the variables presented statistically significant differences”.
- I don’t understand this sentence. I don’t find the T1 adenocarcinoma group in Table 3. Please specify.
On line 245 we have also changed the sentence: “In group II, again, because of low number of patients with uncertain diagnosis, a multivariate analysis was not performed.”
- please correct this in the text, there you write despite instead of because.
- also line 253: please add predictive factors for what?
Author Response
Cancers Journal
Reviewer 1
Comments from authors:
Thank you for your very thorough comments, in response to which we have carried out a revision of our paper.
Comments and Suggestions for Authors
Thanks for revising the manuscript. It’s scientific impact improved. I still have a couple of issues:
- 5. Conclusions: please delete the first part, this is results, not conclusion
We have deleted this first part.
- 5. Conclusions: In the case of… please indicate at the end of the sentence WHY you think this should be the right way to do. Should it be despite instead of in spite?
Thank you. We have rewritten the conclusions.
- 5. Conclusions: Last two sentences. I don’t understand what the authors want to say. What should a reader do in those situations?
We have tried to clarify the sentence:
“Multicenter studies with larger samples ofT2-T3 rectal adenomas and T1-T2 adenocarcinomas with uncertain diagnosis should now be performed to further assess this therapeutic strategy”
- Abbreviation usmr: sorry, this abbreviation is not very intuitive… would consider to change – in the table it is us/mr
We agree with your suggestion. We have changed usmr to us/mr
-How were patients treated with TES but suspicious adenopathies? Nodes were not resected with TES, correct?
All the patients diagnosed with suspicious adenopathies presented adenomas in the biopsy and in the repeat biopsy. The aim in TES is total rectal wall excision without affecting the perirectal fat,,where the nodes are located.
Those two patients had adenoma rectal tumors (table2). The biopsy was repeated. In one patient after TES the definitive pathology was still adenoma (and was considered an inflammatory node). In the other, the definitive pathology after TES was a T1 adenocarcinoma, and so completion surgery with TME was performed.
- I think that this has to be mentioned in the text!
- We have included this in the text.
- Table 1: please provide the number of eligible patients (e.g. MRI) or indicate how many did not have a certain diagnostic
Table 1 presents patients’ demographic characteristics and preoperative data on tumors in groups I and II, and table 3 displays data on diagnostic uncertainty in group I and II patients.
We define uncertain diagnosis in groups I and II in the text (line 143) and also in tables 1 and 2.
Tables 1 and 3 are complementary tables with information on patients’ demographic characteristics and preoperative data of tumors in groups I and II, and the analysis of group I and II patients regarding diagnostic uncertainty
- if not all patients have had a MR exam, this has to be mentioned. Same e.g. for TES after polypectomy, this does not sum up to 529. What happened to the remaining 13 patients? Same for lesion morphology… not summing up numbers is very confusing
In line 127 we mentioned that not all patients underwent MR. This is why, when we add up the patients who underwent MR the total is 277 (232+45).
We have checked all the data and clarifying the variables in which the number does not add up to 529.
- How comes that 40 patients of group 2 did not have a pathology report after TES? Was the specimen not sent to pathology or was no tumor left in the specimen?
You are right: as explained in table 3, these patients did not present pathology after post-polypectomy TES. We discussed these cases in a previous study (reference 14):
- This manuscript has the focus final pathology results. How can you keep patients in the analysis where you don’t have a final pathology report? Please comment.
All the patients had a pathology report but in these patients TES was indicated because of the suspicion of residual adenocarcinoma. However, no adenocarcinoma or adenoma was found
- I don't think it is a fair point mentioning that just a small number of patients is clinically not relevant (e.g. group II), this should be corrected
Thank you for this observation. We have changed the sentence starting on line 230 to: ”In the T1 adenocarcinoma group, despite the low number of patients with uncertain diagnosis, none of the variables presented statistically significant differences”.
- I don’t understand this sentence. I don’t find the T1 adenocarcinoma group in Table 3. Please specify.
The T1 adenocarcinoma group is group II. So to be more clear, we now speak of “group II” in the text.
On line 245 we have also changed the sentence: “In group II, again, because of low number of patients with uncertain diagnosis, a multivariate analysis was not performed.”
- please correct this in the text, there you write despite instead of because.
Thank you, this has been corrected.
- also line 253: please add predictive factors for what?
Predictive factors for adenocarcinoma pT greater than T1
Submission Date
06 June 2021
Date of this review
10 Jul 2021 16:04:15
